# Poincaré Embeddings for Learning Hierarchical Representations

**Maximilian Nickel**
Facebook AI Research
maxn@fb.com

**Douwe Kiela**
Facebook AI Research
dkiela@fb.com

## Abstract

Representation learning has become an invaluable approach for learning from symbolic data such as text and graphs. However, state-of-the-art embedding methods typically do not account for latent hierarchical structures which are characteristic for many complex symbolic datasets. In this work, we introduce a new approach for learning hierarchical representations of symbolic data by embedding them into hyperbolic space – or more precisely into an $n$-dimensional Poincaré ball. Due to the underlying hyperbolic geometry, this allows us to learn parsimonious representations of symbolic data by simultaneously capturing hierarchy and similarity. We present an efficient algorithm to learn the embeddings based on Riemannian optimization and show experimentally that Poincaré embeddings can outperform Euclidean embeddings significantly on data with latent hierarchies, both in terms of representation capacity and in terms of generalization ability.

## 1 Introduction

Learning representations of symbolic data such as text, graphs and multi-relational data has become a central paradigm in machine learning and artificial intelligence. For instance, word embeddings such as WORD2VEC [20], GLOVE [27] and FASTTEXT [5, 16] are widely used for tasks ranging from machine translation to sentiment analysis. Similarly, embeddings of graphs such as latent space embeddings [15], NODE2VEC [13], and DEEPWALK [28] have found important applications for community detection and link prediction in social networks. Furthermore, embeddings of multi-relational data such as RESCAL [22], TRANSE [7], and Universal Schema [31] are being used for knowledge graph completion and information extraction.

Typically, the objective of an embedding method is to organize symbolic objects (e.g., words, entities, concepts) in a way such that their similarity or distance in the embedding space reflects their semantic similarity. For instance, Mikolov et al. [20] embed words in $\mathbb{R}^d$ such that their inner product is maximized when words co-occur within similar contexts in text corpora. This is motivated by the distributional hypothesis [14, 11], i.e., that the meaning of words can be derived from the contexts in which they appear. Similarly, Hoff et al. [15] embed social networks such that the distance between social actors is minimized if they are connected in the network. This reflects the homophily property that is characteristic for many networks, i.e. that similar actors tend to associate with each other.

Although embedding methods have proven successful in numerous applications, they suffer from a fundamental limitation: their ability to model complex patterns is inherently bounded by the dimensionality of the embedding space. For instance, Nickel et al. [23] showed that linear embeddings of graphs can require a prohibitively large dimensionality to model certain types of relations. Although non-linear embeddings can mitigate this problem [8], complex graph patterns can still require a computationally infeasible embedding dimension. As a consequence, no method yet exists that is able to compute embeddings of large graph-structured data – such as social networks, knowledge graphs or taxonomies – without loss of information. Since the ability to express information is a

precondition for learning and generalization, it is therefore important to increase the representation capacity of embedding methods such that they can realistically be used to model complex patterns on a large scale. In this work, we focus on mitigating this problem for a certain class of symbolic data, i.e., large datasets whose objects can be organized according to a latent hierarchy – a property that is inherent in many complex datasets. For instance, the existence of power-law distributions in datasets can often be traced back to hierarchical structures [29]. Prominent examples of power-law distributed data include natural language (Zipf's law [40]) and scale-free networks such as social and semantic networks [32]. Similarly, the empirical analysis of Adcock et al. [1] indicated that many real-world networks exhibit an underlying tree-like structure.

To exploit this structural property for learning more efficient representations, we propose to compute embeddings not in Euclidean but in hyperbolic space, i.e., space with constant negative curvature. Informally, hyperbolic space can be thought of as a continuous version of trees and as such it is naturally equipped to model hierarchical structures. For instance, it has been shown that any finite tree can be embedded into a finite hyperbolic space such that distances are preserved approximately [12]. We base our approach on a particular model of hyperbolic space, i.e., the Poincaré ball model, as it is well-suited for gradient-based optimization. This allows us to develop an efficient algorithm for computing the embeddings based on Riemannian optimization, which is easily parallelizable and scales to large datasets. Experimentally, we show that our approach can provide high quality embeddings of large taxonomies – both with and without missing data. Moreover, we show that embeddings trained on WORDNET provide state-of-the-art performance for lexical entailment. On collaboration networks, we also show that Poincaré embeddings are successful in predicting links in graphs where they outperform Euclidean embeddings, especially in low dimensions.

The remainder of this paper is organized as follows: In Section 2 we briefly review hyperbolic geometry and discuss related work. In Section 3 we introduce Poincaré embeddings and present a scalable algorithm to compute them. In Section 4 we evaluate our approach on tasks such as taxonomy embedding, link prediction in networks and predicting lexical entailment.

## 2 Embeddings and Hyperbolic Geometry

Hyperbolic geometry is a non-Euclidean geometry which studies spaces of constant negative curvature. It is, for instance, related to Minkowski spacetime in special relativity. In network science, hyperbolic spaces have started to receive attention as they are well-suited to model hierarchical data. For instance, consider the task of embedding a tree into a metric space such that its structure is reflected in the embedding. A regular tree with branching factor $b$ has $(b + 1)b^{\ell-1}$ nodes at level $\ell$ and $((b + 1)b^{\ell} - 2)/(b - 1)$ nodes on a level less or equal than $\ell$. Hence, the number of children grows exponentially with their distance to the root of the tree. In hyperbolic geometry this kind of tree structure can be modeled easily in two dimensions: nodes that are *exactly* $\ell$ levels below the root are placed on a sphere in hyperbolic space with radius $r \propto \ell$ and nodes that are *less than* $\ell$ levels below the root are located within this sphere. This type of construction is possible as hyperbolic disc area and circle length grow exponentially with their radius.[1] See Figure 1b for an example. Intuitively, hyperbolic spaces can be thought of as continuous versions of trees or vice versa, trees can be thought of as "discrete hyperbolic spaces" [19]. In $\mathbb{R}^2$, a similar construction is not possible as circle length ($2\pi r$) and disc area ($2\pi r^2$) grow only linearly and quadratically with regard to $r$ in Euclidean geometry. Instead, it is necessary to increase the dimensionality of the embedding to model increasingly complex hierarchies. As the number of parameters increases, this can lead to computational problems in terms of runtime and memory complexity as well as to overfitting.

Due to these properties, hyperbolic space has recently been considered to model complex networks. For instance, Kleinberg [18] introduced hyperbolic geometry for greedy routing in geographic communication networks. Similarly, Boguñá et al. [4] proposed hyperbolic embeddings of the AS Internet topology to perform greedy shortest path routing in the embedding space. Krioukov et al. [19] developed a geometric framework to model complex networks using hyperbolic space and showed how typical properties such as heterogeneous degree distributions and strong clustering can emerge by assuming an underlying hyperbolic geometry to networks. Furthermore, Adcock et al.

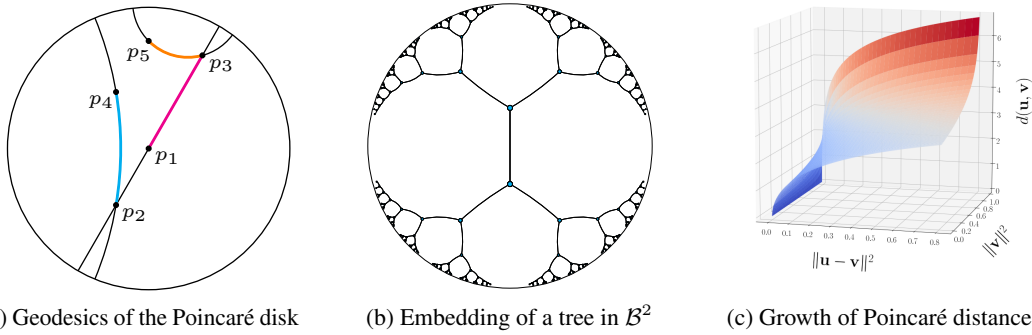

(a) Geodesics of the Poincaré disk   (b) Embedding of a tree in $\mathcal{B}^2$   (c) Growth of Poincaré distance

Figure 1: (a) Due to the negative curvature of $\mathcal{B}$, the distance of points increases exponentially (relative to their Euclidean distance) the closer they are to the boundary. (c) Growth of the Poincaré distance $d(\boldsymbol{u}, \boldsymbol{v})$ relative to the Euclidean distance and the norm of $\boldsymbol{v}$ (for fixed $\|\boldsymbol{u}\| = 0.9$). (b) Embedding of a regular tree in $\mathcal{B}^2$ such that all connected nodes are spaced equally far apart (i.e., all black line segments have identical hyperbolic length).

[1] proposed a measure based on Gromov's $\delta$-hyperbolicity [12] to characterize the tree-likeness of graphs. Ontrup and Ritter [25] proposed hyperbolic self-organizing maps for data exploration. Asta and Shalizi [3] used hyperbolic embeddings to compare the global structure of networks. Sun et al. [33] proposed Space-Time embeddings to learn representations of non-metric data.

Euclidean embeddings, on the other hand, have become a popular approach to represent symbolic data in machine learning and artificial intelligence. For instance, in addition to the methods discussed in Section 1, Paccanaro and Hinton [26] proposed one of the first embedding methods to learn from relational data. More recently, Holographic [24] and Complex Embeddings [34] have shown state-of-the-art performance in Knowledge Graph completion. In relation to hierarchical representations, Vilnis and McCallum [36] proposed to learn density-based word representations, i.e., Gaussian embeddings, to capture uncertainty and asymmetry. Given ordered input pairs, Vendrov et al. [35] proposed Order Embeddings to model visual-semantic hierarchies over words, sentences, and images. Demeester et al. [10] showed that including prior information about hypernymy relations in form of logical rules can improve the quality of word embeddings.

## 3   Poincaré Embeddings

In the following, we are interested in finding embeddings of symbolic data such that their distance in the embedding space reflects their semantic similarity. We assume that there exists a latent hierarchy in which the symbols can be organized. In addition to the similarity of objects, we intend to also reflect this hierarchy in the embedding space to improve over existing methods in two ways:

1. By inducing an appropriate structural bias on the embedding space we aim at improving generalization performance as well as runtime and memory complexity.

2. By capturing the hierarchy explicitly in the embedding space, we aim at gaining additional insights about the relationships between symbols and the importance of individual symbols.

Although we assume that there exists a latent hierarchy, we do not assume that we have direct access to information about this hierarchy, e.g., via ordered input pairs. Instead, we consider the task of inferring the hierarchical relationships fully unsupervised, as is, for instance, necessary for text and network data. For these reasons – and motivated by the discussion in Section 2 – we embed symbolic data into hyperbolic space $\mathbb{H}$. In contrast to Euclidean space $\mathbb{R}$, there exist multiple, equivalent models of $\mathbb{H}$ such as the Beltrami-Klein model, the hyperboloid model, and the Poincaré half-plane model. In the following, we will base our approach on the Poincaré ball model, as it is well-suited for gradient-based optimization. In particular, let $\mathcal{B}^d = \{\boldsymbol{x} \in \mathbb{R}^d \mid \|\boldsymbol{x}\| < 1\}$ be the *open* $d$-dimensional unit ball, where $\|\cdot\|$ denotes the Euclidean norm. The Poincaré ball model of hyperbolic space corresponds then to the Riemannian manifold $(\mathcal{B}^d, g_{\boldsymbol{x}})$, i.e., the open unit ball equipped with the Riemannian metric tensor

$$g_{\boldsymbol{x}} = \left( \frac{2}{1 - \|\boldsymbol{x}\|^2} \right)^2 g^E,$$

where $\boldsymbol{x} \in \mathcal{B}^d$ and $g^E$ denotes the Euclidean metric tensor. Furthermore, the distance between points $\boldsymbol{u}, \boldsymbol{v} \in \mathcal{B}^d$ is given as

$$d(\boldsymbol{u}, \boldsymbol{v}) = \text{arcosh}\left(1 + 2\frac{\|\boldsymbol{u} - \boldsymbol{v}\|^2}{(1 - \|\boldsymbol{u}\|^2)(1 - \|\boldsymbol{v}\|^2)}\right). \tag{1}$$

The boundary of the ball is denoted by $\partial \mathcal{B}$. It corresponds to the sphere $\mathcal{S}^{d-1}$ and is not part of the manifold, but represents infinitely distant points. Geodesics in $\mathcal{B}^d$ are then circles that are orthogonal to $\partial \mathcal{B}$ (as well as all diameters). See Figure 1a for an illustration.

It can be seen from Equation (1), that the distance within the Poincaré ball changes smoothly with respect to the location of $\boldsymbol{u}$ and $\boldsymbol{v}$. This locality property of the Poincaré distance is key for finding continuous embeddings of hierarchies. For instance, by placing the root node of a tree at the origin of $\mathcal{B}^d$ it would have a relatively small distance to all other nodes as its Euclidean norm is zero. On the other hand, leaf nodes can be placed close to the boundary of the Poincaré ball as the distance grows very fast between points with a norm close to one. Furthermore, please note that Equation (1) is symmetric and that the hierarchical organization of the space is solely determined by the distance of points to the origin. Due to this self-organizing property, Equation (1) is applicable in an unsupervised setting where the hierarchical order of objects is not specified in advance such as text and networks. Remarkably, Equation (1) allows us therefore to learn embeddings that simultaneously capture the hierarchy of objects (through their norm) as well a their similarity (through their distance).

Since a single hierarchical structure can be well represented in two dimensions, the Poincaré disk ($\mathcal{B}^2$) is a common way to model hyperbolic geometry. In our method, we instead use the Poincaré ball ($\mathcal{B}^d$), for two main reasons: First, in many datasets such as text corpora, multiple latent hierarchies can co-exist, which can not always be modeled in two dimensions. Second, a larger embedding dimension can decrease the difficulty for an optimization method to find a good embedding (also for single hierarchies) as it allows for more degrees of freedom during the optimization process.

To compute Poincaré embeddings for a set of symbols $\mathcal{S} = \{x_i\}_{i=1}^n$, we are then interested in finding embeddings $\Theta = \{\boldsymbol{\theta}_i\}_{i=1}^n$, where $\boldsymbol{\theta}_i \in \mathcal{B}^d$. We assume we are given a problem-specific loss function $\mathcal{L}(\Theta)$ which encourages semantically similar objects to be close in the embedding space according to their Poincaré distance. To estimate $\Theta$, we then solve the optimization problem

$$\Theta' \leftarrow \underset{\Theta}{\arg\min}\, \mathcal{L}(\Theta) \qquad \text{s.t. } \forall\, \boldsymbol{\theta}_i \in \Theta : \|\boldsymbol{\theta}_i\| < 1. \tag{2}$$

We will discuss specific loss functions in Section 4.

### 3.1 Optimization

Since the Poincaré Ball has a Riemannian manifold structure, we can optimize Equation (2) via stochastic Riemannian optimization methods such as RSGD [6] or RSVRG [39]. In particular, let $\mathcal{T}_\theta \mathcal{B}$ denote the tangent space of a point $\boldsymbol{\theta} \in \mathcal{B}^d$. Furthermore, let $\nabla_R \in \mathcal{T}_\theta \mathcal{B}$ denote the Riemannian gradient of $\mathcal{L}(\boldsymbol{\theta})$ and let $\nabla_E$ denote the Euclidean gradient of $\mathcal{L}(\boldsymbol{\theta})$. Using RSGD, parameter updates to minimize Equation (2) are then of the form

$$\boldsymbol{\theta}_{t+1} = \mathfrak{R}_{\theta_t}\left(-\eta_t \nabla_R \mathcal{L}(\boldsymbol{\theta}_t)\right)$$

where $\mathfrak{R}_{\theta_t}$ denotes the retraction onto $\mathcal{B}$ at $\boldsymbol{\theta}$ and $\eta_t$ denotes the learning rate at time $t$. Hence, for the minimization of Equation (2), we require the Riemannian gradient and a suitable retraction. Since the Poincaré ball is a conformal model of hyperbolic space, the angles between adjacent vectors are identical to their angles in the Euclidean space. The length of vectors however might differ. To derive the Riemannian gradient from the Euclidean gradient, it is sufficient to rescale $\nabla_E$ with the inverse of the Poincaré ball metric tensor, i.e., $g_\theta^{-1}$. Since $g_\theta$ is a scalar matrix, the inverse is trivial to compute. Furthermore, since Equation (1) is fully differentiable, the Euclidean gradient can easily be derived using standard calculus. In particular, the Euclidean gradient $\nabla_E = \frac{\partial \mathcal{L}(\boldsymbol{\theta})}{\partial d(\boldsymbol{\theta}, \boldsymbol{x})}\frac{\partial d(\boldsymbol{\theta}, \boldsymbol{x})}{\partial \boldsymbol{\theta}}$ depends on the gradient of $\mathcal{L}$, which we assume is known, and the partial derivatives of the Poincaré distance, which can be computed as follows: Let $\alpha = 1 - \|\boldsymbol{\theta}\|^2$, $\beta = 1 - \|\boldsymbol{x}\|^2$ and let $\gamma = 1 + \frac{2}{\alpha\beta}\|\boldsymbol{\theta} - \boldsymbol{x}\|^2$. The partial derivate of the Poincaré distance with respect to $\boldsymbol{\theta}$ is then given as

$$\frac{\partial d(\boldsymbol{\theta}, \boldsymbol{x})}{\partial \boldsymbol{\theta}} = \frac{4}{\beta\sqrt{\gamma^2 - 1}}\left(\frac{\|\boldsymbol{x}\|^2 - 2\langle\boldsymbol{\theta}, \boldsymbol{x}\rangle + 1}{\alpha^2}\boldsymbol{\theta} - \frac{\boldsymbol{x}}{\alpha}\right). \tag{3}$$

Since $d(\cdot, \cdot)$ is symmetric, the partial derivative $\frac{\partial d(\boldsymbol{x}, \boldsymbol{\theta})}{\partial \boldsymbol{\theta}}$ can be derived analogously. As retraction operation we use $\mathfrak{R}_\theta(\boldsymbol{v}) = \boldsymbol{\theta} + \boldsymbol{v}$. In combination with the Riemannian gradient, this corresponds then to the well-known natural gradient method [2]. Furthermore, we constrain the embeddings to remain within the Poincaré ball via the projection

$$\text{proj}(\boldsymbol{\theta}) = \begin{cases} \boldsymbol{\theta}/\|\boldsymbol{\theta}\| - \varepsilon & \text{if } \|\boldsymbol{\theta}\| \geq 1 \\ \boldsymbol{\theta} & \text{otherwise} , \end{cases}$$

where $\varepsilon$ is a small constant to ensure numerical stability. In all experiments we used $\varepsilon = 10^{-5}$. In summary, the full update for a single embedding is then of the form

$$\boldsymbol{\theta}_{t+1} \leftarrow \text{proj}\left(\boldsymbol{\theta}_t - \eta_t \frac{(1 - \|\boldsymbol{\theta}_t\|^2)^2}{4} \nabla_E\right). \tag{4}$$

It can be seen from Equations (3) and (4) that this algorithm scales well to large datasets, as the computational and memory complexity of an update depends linearly on the embedding dimension. Moreover, the algorithm is straightforward to parallelize via methods such as Hogwild [30], as the updates are sparse (only a small number of embeddings are modified in an update) and collisions are very unlikely on large-scale data.

## 3.2 Training Details

In addition to this optimization procedure, we found that the following training details were helpful for obtaining good representations: First, we initialize all embeddings randomly from the uniform distribution $\mathcal{U}(-0.001, 0.001)$. This causes embeddings to be initialized close to the origin of $\mathcal{B}^d$. Second, we found that a good initial angular layout can be helpful to find good embeddings. For this reason, we train during an initial "burn-in" phase with a reduced learning rate $\eta/c$. In combination with initializing close to the origin, this can improve the angular layout without moving too far towards the boundary. In our experiments, we set $c = 10$ and the duration of the burn-in to 10 epochs.

## 4 Evaluation

In this section, we evaluate the quality of Poincaré embeddings for a variety of tasks, i.e., for the embedding of taxonomies, for link prediction in networks, and for modeling lexical entailment. In all tasks, we train on data where the hierarchy of objects is not explicitly encoded. This allows us to evaluate the ability of the embeddings to infer hierachical relationships without supervision. Moreover, since we are mostly interested in the properties of the metric space, we focus on embeddings based purely on the Poincaré distance and on models with comparable expressivity. In particular, we compare the **Poincaré** distance as defined in Equation (1) to the following two distance functions:

**Euclidean** In all cases, we include the Euclidean distance $d(\boldsymbol{u}, \boldsymbol{v}) = \|\boldsymbol{u} - \boldsymbol{v}\|^2$. As the Euclidean distance is flat and symmetric, we expect that it requires a large dimensionality to model the hierarchical structure of the data.

**Translational** For asymmetric data, we also include the score function $d(\boldsymbol{u}, \boldsymbol{v}) = \|\boldsymbol{u} - \boldsymbol{v} + \boldsymbol{r}\|^2$, as proposed by Bordes et al. [7] for modeling large-scale graph-structured data. For this score function, we also learn the global translation vector $\boldsymbol{r}$ during training.

Note that the translational score function has, due to its asymmetry, more information about the nature of an embedding problem than a symmetric distance when the order of $(u, v)$ indicates the hierarchy of elements. This is, for instance, the case for $\texttt{is-a}(u, v)$ relations in taxonomies. For the Poincaré distance and the Euclidean distance we could randomly permute the order of $(u, v)$ and obtain the identical embedding, while this is not the case for the translational score function. As such, it is not fully unsupervised and only applicable where this hierarchical information is available.

### 4.1 Embedding Taxonomies

In the first set of experiments, we are interested in evaluating the ability of Poincaré embeddings to embed data that exhibits a clear latent hierarchical structure. For this purpose, we conduct experiments on the *transitive closure* of the WORDNET noun hierarchy [21] in two settings:

Table 1: Experimental results on the transitive closure of the WORDNET noun hierarchy. Highlighted cells indicate the best Euclidean embeddings as well as the Poincaré embeddings which achieve equal or better results. Bold numbers indicate absolute best results.

| | | | **Dimensionality** | | | | | |
| | | | 5 | 10 | 20 | 50 | 100 | 200 |
|---|---|---|---|---|---|---|---|---|
| **WORDNET Reconstruction** | **Euclidean** | Rank | 3542.3 | 2286.9 | 1685.9 | 1281.7 | 1187.3 | 1157.3 |
| | | MAP | 0.024 | 0.059 | 0.087 | 0.140 | 0.162 | 0.168 |
| | **Translational** | Rank | 205.9 | 179.4 | 95.3 | 92.8 | 92.7 | 91.0 |
| | | MAP | 0.517 | 0.503 | 0.563 | 0.566 | 0.562 | 0.565 |
| | **Poincaré** | Rank | 4.9 | 4.02 | 3.84 | 3.98 | 3.9 | **3.83** |
| | | MAP | 0.823 | 0.851 | 0.855 | 0.86 | 0.857 | **0.87** |
| **WORDNET Link Pred.** | **Euclidean** | Rank | 3311.1 | 2199.5 | 952.3 | 351.4 | 190.7 | 81.5 |
| | | MAP | 0.024 | 0.059 | 0.176 | 0.286 | 0.428 | 0.490 |
| | **Translational** | Rank | 65.7 | 56.6 | 52.1 | 47.2 | 43.2 | 40.4 |
| | | MAP | 0.545 | 0.554 | 0.554 | 0.56 | 0.562 | 0.559 |
| | **Poincaré** | Rank | 5.7 | **4.3** | 4.9 | 4.6 | 4.6 | 4.6 |
| | | MAP | 0.825 | 0.852 | 0.861 | **0.863** | 0.856 | 0.855 |

**Reconstruction** To evaluate representation capacity, we embed fully observed data and reconstruct it from the embedding. The reconstruction error in relation to the embedding dimension is then a measure for the capacity of the model.

**Link Prediction** To test generalization performance, we split the data into a train, validation and test set by randomly holding out observed links. The validation and test set do not include links involving root or leaf nodes as these links would either be trivial or impossible to predict reliably.

Since we are embedding the transitive closure, the hierarchical structure is not directly visible from the raw data but has to be inferred. For Poincaré and Euclidean embeddings we additionaly remove the directionality of the edges and embed undirected graphs. The transitive closure of the WORDNET noun hierarchy consists of 82,115 nouns and 743,241 hypernymy relations.

On this data, we learn embeddings in both settings as follows: Let $\mathcal{D} = \{(u, v)\}$ be the set of observed hypernymy relations between noun pairs. We then learn embeddings of all symbols in $\mathcal{D}$ such that related objects are close in the embedding space. In particular, we minimize the loss function

$$\mathcal{L}(\Theta) = \sum_{(u,v) \in \mathcal{D}} \log \frac{e^{-d(\boldsymbol{u},\boldsymbol{v})}}{\sum_{\boldsymbol{v'} \in \mathcal{N}(u)} e^{-d(\boldsymbol{u},\boldsymbol{v'})}}, \qquad (5)$$

where $\mathcal{N}(u) = \{v' \mid (u, v') \notin \mathcal{D}\} \cup \{v\}$ is the set of negative examples for $u$ (including $v$). For training, we randomly sample 10 negative examples per positive example. Equation (5) is similar to the loss used in Linear Relational Embeddings [26] (with additional negative sampling) and encourages related objects to be closer to each other than objects for which we didn't observe a relationship. This choice of loss function is motivated by the observation that we don't want to push symbols that belong to distinct subtrees arbitrarily far apart, as their subtrees might still be close. Instead, we only want them to be farther apart than symbols with an observed relation.

We evaluate the quality of the embeddings as commonly done for graph embeddings [7, 24]: For each observed relationship $(u, v)$, we rank its distance $d(\boldsymbol{u}, \boldsymbol{v})$ among the ground-truth negative examples for $u$, i.e., among the set $\{d(\boldsymbol{u}, \boldsymbol{v'}) \mid (u, v') \notin \mathcal{D}\}$. In the Reconstruction setting, we evaluate the ranking on all nouns in the dataset. We then record the mean rank of $v$ as well as the mean average precision (MAP) of the ranking. The results of these experiments are shown in Table 1. It can be seen that Poincaré embeddings are very successful in the embedding of large taxonomies – both with regard to their representation capacity and their generalization performance. Even compared to Translational embeddings, which have more information about the structure of the task, Poincaré embeddings show a greatly improved performance while using an embedding that is smaller by an order of magnitude. Furthermore, the results of Poincaré embeddings in the link prediction task are robust with regard to the embedding dimension. We attribute this result to the structural bias of

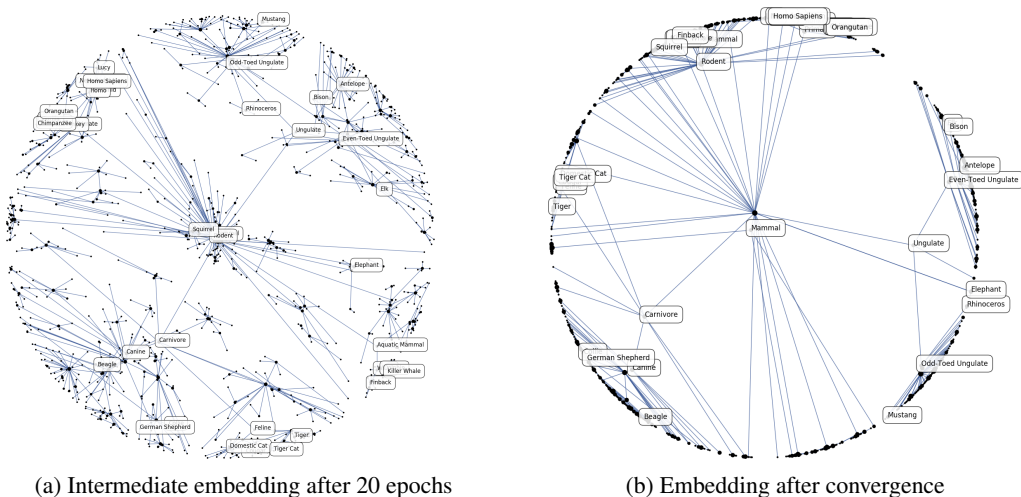

| (a) Intermediate embedding after 20 epochs | (b) Embedding after convergence |

Figure 2: Two-dimensional Poincaré embeddings of transitive closure of the WORDNET mammals subtree. Ground-truth `is-a` relations of the original WORDNET tree are indicated via blue edges. A Poincaré embedding with $d = 5$ achieves mean rank 1.26 and MAP 0.927 on this subtree.

the embedding space which could lead to reduced overfitting on data with a clear latent hierarchy. Additionally, Figure 2 shows a visualization of a two-dimensional Poincaré embedding. For the purpose of clarity, this embedding has been trained only on the mammals subtree of WORDNET.

## 4.2 Network Embeddings

Next, we evaluated the performance of Poincaré embeddings for modeling complex networks. Since edges in such networks can often be explained via latent hierarchies over their nodes [9], we are interested in the benefits of Poincaré embeddings in terms representation size and generalization performance. We performed our experiments on four commonly used social networks, i.e, ASTROPH, CONDMAT, GRQC, and HEPPH. These networks represent scientific collaborations such that there exists an undirected edge between two persons if they co-authored a paper. For these networks, we model the probability of an edge as proposed by Krioukov et al. [19] via the Fermi-Dirac distribution

$$P((u, v) = 1 \mid \Theta) = \frac{1}{e^{(d(\boldsymbol{u}, \boldsymbol{v}) - r)/t} + 1} \tag{6}$$

where $r, t > 0$ are hyperparameters. Here, $r$ corresponds to the radius around each point $\boldsymbol{u}$ such that points within this radius are likely to have an edge with $u$. The parameter $t$ specifies the steepness of the logistic function and influences both average clustering as well as the degree distribution [19]. We use the cross-entropy loss to learn the embeddings and sample negatives as in Section 4.1.

For evaluation, we split each dataset randomly into train, validation, and test set. The hyperparameters $r$ and $t$ were tuned for each method on the validation set. Table 2 lists the MAP score of Poincaré and Euclidean embeddings on the test set for the hyperparameters with the best validation score. Additionally, we also list the reconstruction performance without missing data. Translational embeddings are not applicable to these datasets as they consist of undirected edges. It can be seen that Poincaré embeddings perform again very well on these datasets and – especially in the low-dimensional regime – outperform Euclidean embeddings.

## 4.3 Lexical Entailment

An interesting aspect of Poincaré embeddings is that they allow us to make graded assertions about hierarchical relationships, as hierarchies are represented in a continuous space. We test this property on HYPERLEX [37], which is a gold standard resource for evaluating how well semantic models capture graded lexical entailment by quantifying to what *degree X* is a type of $Y$ via ratings on a scale of $[0, 10]$. Using the noun part of HYPERLEX, which consists of 2163 rated noun pairs, we then evaluated how well Poincaré embeddings reflect these graded assertions. For this purpose, we

Table 2: Mean average precision for Reconstruction and Link Prediction on network data.

| | | Dimensionality | | | | | | | |
| --- | --- | --- | --- | --- | --- | --- | --- | --- | --- |
| | | Reconstruction | | | | Link Prediction | | | |
| | | 10 | 20 | 50 | 100 | 10 | 20 | 50 | 100 |
| ASTROPH | **Euclidean** | 0.376 | 0.788 | 0.969 | 0.989 | 0.508 | 0.815 | 0.946 | 0.960 |
| N=18,772; E=198,110 | **Poincaré** | 0.703 | 0.897 | 0.982 | 0.990 | 0.671 | 0.860 | 0.977 | 0.988 |
| CONDMAT | **Euclidean** | 0.356 | 0.860 | 0.991 | 0.998 | 0.308 | 0.617 | 0.725 | 0.736 |
| N=23,133; E=93,497 | **Poincaré** | 0.799 | 0.963 | 0.996 | 0.998 | 0.539 | 0.718 | 0.756 | 0.758 |
| GRQC | **Euclidean** | 0.522 | 0.931 | 0.994 | 0.998 | 0.438 | 0.584 | 0.673 | 0.683 |
| N=5,242; E=14,496 | **Poincaré** | 0.990 | 0.999 | 0.999 | 0.999 | 0.660 | 0.691 | 0.695 | 0.697 |
| HEPPH | **Euclidean** | 0.434 | 0.742 | 0.937 | 0.966 | 0.642 | 0.749 | 0.779 | 0.783 |
| N=12,008; E=118,521 | **Poincaré** | 0.811 | 0.960 | 0.994 | 0.997 | 0.683 | 0.743 | 0.770 | 0.774 |

Table 3: Spearman's $\rho$ for Lexical Entailment on HYPERLEX.

| | FR | SLQS-Sim | WN-Basic | WN-WuP | WN-LCh | Vis-ID | Euclidean | Poincaré |
| --- | --- | --- | --- | --- | --- | --- | --- | --- |
| $\rho$ | 0.283 | 0.229 | 0.240 | 0.214 | 0.214 | 0.253 | 0.389 | 0.512 |

used the Poincaré embeddings that were obtained in Section 4.1 by embedding WORDNET with a dimensionality $d = 5$. Note that these embeddings were not specifically trained for this task. To determine to what extent `is-a`$(u, v)$ is true, we used the score function:

$$\text{score}(\text{is-a}(u, v)) = -(1 + \alpha(\|\boldsymbol{v}\| - \|\boldsymbol{u}\|))d(\boldsymbol{u}, \boldsymbol{v}). \tag{7}$$

Here, the term $\alpha(\|\boldsymbol{v}\| - \|\boldsymbol{u}\|)$ acts as a penalty when $v$ is lower in the embedding hierarchy, i.e., when $\boldsymbol{v}$ has a higher norm than $\boldsymbol{u}$. The hyperparameter $\alpha$ determines the severity of the penalty. In our experiments we set $\alpha = 10^3$.

Using Equation (7), we scored all noun pairs in HYPERLEX and recorded Spearman's rank correlation with the ground-truth ranking. The results of this experiment are shown in Table 3. It can be seen that the ranking based on Poincaré embeddings clearly outperforms all state-of-the-art methods evaluated in [37]. Methods in Table 3 that are prefixed with WN also use WORDNET as a basis and therefore are most comparable. The same embeddings also achieved a state-of-the-art accuracy of $0.86$ on WBLESS [38, 17], which evaluates non-graded lexical entailment.

## 5 Discussion and Future Work

In this paper, we introduced Poincaré embeddings for learning representations of symbolic data and showed how they can simultaneously learn the similarity and the hierarchy of objects. Furthermore, we proposed an efficient algorithm to compute the embeddings and showed experimentally, that Poincaré embeddings provide important advantages over Euclidean embeddings on hierarchical data: First, Poincaré embeddings enable parsimonious representations that allow us to learn high-quality embeddings of large-scale taxonomies. Second, excellent link prediction results indicate that hyperbolic geometry can introduce an important structural bias for the embedding of complex symbolic data. Third, state-of-the-art results for predicting lexical entailment suggest that the hierarchy in the embedding space corresponds well to the underlying semantics of the data.

The focus of this work was to evaluate general properties of hyperbolic geometry for the embedding of symbolic data. In future work, we intend to expand the applications of Poincaré embeddings – for instance to multi-relational data – and to derive models that are tailored to specific tasks such as word embeddings. Furthermore, we have shown that natural gradient based optimization already produces very good embeddings and scales to large datasets. We expect that a full Riemannian optimization approach can further increase the quality of the embeddings and lead to faster convergence.

An important aspect of future work regards also the applicability of hyperbolic embeddings in downstream tasks: models that operate on embeddings often make an implicit Euclidean assumption and likely require some adaptation to be compatible with hyperbolic spaces.

## Footnotes

[1] For instance, in a two dimensional hyperbolic space with constant curvature $K = -1$, the length of a circle is given as $2\pi \sinh r$ while the area of a disc is given as $2\pi(\cosh r - 1)$. Since $\sinh r = \frac{1}{2}(e^r - e^{-r})$ and $\cosh r = \frac{1}{2}(e^r + e^{-r})$, both disc area and circle length grow exponentially with $r$.

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
