[Reviews · NeurIPS 2017]

Reviewer 1



The authors propose an embedding method which learns distributed representations in hyperbolic space rather than Euclidean space. As a result the embeddings capture not only semantic, but also hierarchical structure. The authors introduce the Poincare embeddings and explain how to fit them to data using a stochastic gradient method appropriate for hyperbolic space. In an experimental section, the Poincare embeddings are evaluated on embedding taxonomies, embedding a network and embedding lexical entailment. In all of these applications the hierarchical nature of the data is encoded in the training data. The abstract, introduction and model section made it seem like it would be possible to learn the hierarchical structure of words from standard text corpora which do not have any relational annotations. Maybe I misunderstood. Is there a way to learn a taxonomy of the vocabulary from unstructured text corpora (say from the Gigaword collection) using Poincare embeddings? How would you choose the loss L in that case? Also, what is the loss L in Section 4.3? The paper is well written, technically interesting, and has interesting results.

Reviewer 2



This paper proposes Poincare embeddings for learning representations of symbolic data. The new method represents symbolic data in an n-dimensional Poincare ball, and can simultaneously learn the similarity and the hierarchy of objects. An algorithm based on Riemannian optimization is introduced to learn these Poincare embeddings. Experimental results show that the Poincare distance outperforms the Euclidean distance and translational distance in tasks like taxonomy embedding, network embedding, and lexical entailment. This paper is generally well-written. The method is clearly presented. The idea seems interesting and might be of potential impact for future work. My only concern is that the empirical evaluation appears to be a bit weak. In each of the tasks evaluated, the authors design a specific learning paradigm for this task (e.g., the loss function of eq.(6) in taxonomy embedding and the edge probability of eq.(7) in network embedding). All the methods use the same paradigm but with different distance metrics (i.e., Poincare distance, Euclidean distance, and translational distance). But this specially designed paradigm (e.g., the edge probability of eq.(7) in network embedding) could be particularly advantageous to the Poincare distance while disadvantageous to the baselines. Comparing with state-of-the-art baselines (in their original forms) will significantly strengthen the evaluation, e.g., DeepWalk[1] and LINE[2] in network embedding. For your reference: [1] Perozzi et al. DeepWalk: Online learning of social representations. In KDD'14. [2] Tang et al. Line: Large-scale information network embedding. In WWW'15.

Reviewer 3



Summary ======= The paper proposes a link prediction model that embeds symbols in a hyperbolic space using Poincaré embeddings. In this space, tree structures can more easily be represented as the distance to points increases exponentially w.r.t. Euclidean distance. The paper is motivated and written well. Furthermore, the presented method is intriguing and I believe it will have a notable impact on link prediction research. My concerns are regarding the comparison to state-of-the-art link prediction and how the method performs if the assumption about a hierarchy in the data is dropped. Strengths ========= - Impressive results for embedding hierarchies in a non-Euclidean space using a much smaller dimension. Weaknesses ========== - Maybe I am missing a point, but why is there no comparison with state-of-the-art neural link prediction models (e.g. HolE/ComplEx)? This would strengthen the experimental evaluation a lot as TransE has consistently be outperformed by these methods. Furthermore, how does Poincaré compare to TransE in terms of runtime? - I would have liked to see more experimental results of the limitations of the proposed method. Given that many real-world datasets have less pronounced hierarchies than WordNet, it would have been nice to show what happens if a hierarchy of symbols cannot be assumed for a large portion of the data. For instance, how would the method perform on FB15k? Generally, I would like to see a more thorough discussion about when Poincaré embeddings can be used in practice and when not. For instance, the authors mention word2vec, GloVe and FastText in the introduction and I guess that Poincaré embeddings cannot serve as drop-in replacement for word vectors in these models as we don't know how to perform certain translations in hyperbolic space or how to apply backprop? Such limitations should be clarified in the paper. Minor Comments ============== - Embedding hierarchies in Euclidean has been attempted in prior work that should be mentioned: - For textual entailment and images: Ivan Vendrov, Ryan Kiros, Sanja Fidler, Raquel Urtasun. Order-Embeddings of Images and Language. ICLR 2016 - For link prediction: Thomas Demeester, Tim Rocktäschel, Sebastian Riedel. Lifted Rule Injection for Relation Embeddings. EMNLP 2016 - Footnote 2: "would be more more difficult" -> "would be more difficult" - L277: "representations whats allow us to" -> "representations that allow us to"